biomaterials/materials science

citral, bamboo, mould, antibacterial mechanism, antibacterial property, mildew proofing

**Author for correspondence:**
Chungui Du
e-mail: chunguidu@163.com

# Inhibition mechanism and antibacterial activity of natural antibacterial agent citral on bamboo mould and its anti-mildew effect on bamboo

Jingjing Zhang, Chungui Du, Qi Li, Ailian Hu, Rui Peng, Fangli Sun and Weigang Zhang

School of Engineering, Zhejiang A&F University, Hangzhou 311300, People's Republic of China

JZ, 0000-0001-6857-0999; CD, 0000-0002-6444-5725;
QL, 0000-0003-3714-182X; AH, 0000-0002-0745-6866;
RP, 0000-0002-8774-7582

Bamboo, a natural material, has been widely used in the fields of decoration, architecture and furniture. However, bamboo is easy to mildew and lose its use value. In this paper, the inhibition mechanism and antibacterial activity of a natural antibacterial agent citral on bamboo mould and its anti-mildew effect on bamboo were studied. The results showed that citral could change the shape of mycelium, destroy the integrity of mycelium structure, cell wall and cell membrane structure, thereby causing leakage of nucleic acids, proteins and other substances in the cell, as well as destroy the pH balance of the inside and outside of the cell, to inhibit or kill mould. When the concentration of citral is $100 \, \text{mg ml}^{-1}$, the antibacterial rates of citral against *Penicillium citrinum* (PC), *Trichoderma viride* (TV), *Aspergillus niger* (AN) and a hybrid fungi group comprising PC, TV and AN (Hun) were more than 100%. However, compared with the direct effect of citral on mould, the antibacterial property of bamboo treated with citral was significantly reduced, the mildew proof effect can be achieved only if the concentration of citral to treat bamboo is increased to more than twice the concentration of citral directly acting on mould.

## 1. Introduction

The increasing demand for the limited forest resources in various applications, has led to the shortage in wood supply [1]. Thus, there is an urgent need to look for new materials as alternatives to wood. Bamboo is one of the fastest-growing natural plants in the world, which grows to its maximum height in about three

months and reaches maturity in 3–4 years [2], well exceeding the 20–60 years growth cycle of traditional timber used in structural applications. Bamboo also has the characteristics of one-time afforestation, and sustainable use without destroying the ecological environment [3]. Out of many natural materials, bamboo offers significant structural and environmental advantages given its rapid growth, moisture resistance, climate tolerance and tensile strength comparable to mild steel, good toughness, low processing cost, biodegradability and so on [4]. Therefore, bamboo and bamboo products have been widely used in the fields of decoration, architecture, furniture, gardens as alternatives to wood [5–8], and there is a momentum of rapid growth. However, bamboo is easier to mildew because it contains more sugar, starch, protein etc. than wood, and mildew causes surface contamination of bamboo, showing brown or black, which causes bamboo to lose its value [9,10], which accounts for about 10% of the bamboo output [11,12]. Thus, it is imperative to prevent mildew in bamboo. At present, physical and chemical treatment methods are commonly used in the anti-mildew treatment of bamboo [13]. Owing to the poor anti-mould effect of physical methods, chemical methods dominate. However, the use of some anti-mildew agents in the chemical method such as sulfur, pentachlorophenol and copper, chromium, arsenic and so on, will cause certain negative effects on the environment and human health [9]. Therefore, there is an urgent need to explore and develop anti-mildew agents with good anti-mildew effects and good environmental performance to solve the current problems faced by bamboo mould-proofing.

In recent years, natural antimicrobials from plant sources have gradually become a hotspot in the field of antimicrobial research because of strong antibacterial activity, broad antibacterial spectrum and they are environment friendly [14]. In particular, citral, mainly derived from the essential oil of *Litsea cubeba*, is attracting considerable research interest in research scholars owing to its strong antibacterial property [15–18] and is expected to be popularized and applied as a natural bamboo anti-mildew agent. However, the use of citral as an anti-mildew agent for bamboo is confronted with many problems and challenges.

First, does citral have a strong inhibitory effect on bamboo mould? What is the inhibition mechanism? It has not been elucidated. Second, the moulds that cause bamboo mildew mainly includes *Penicillium citrinum* (PC), *Trichoderma viride* (TV) and *Aspergillus niger* (AN), however, bamboo mildew is usually the result of the combined effects of these three moulds; even if citral affects these three moulds each has a strong inhibitory effect, but is there still a strong inhibitory effect on their mixed moulds? It is still unknown. Thirdly, citral is very unstable during storage and is easily oxidized and volatilized to lose its antibacterial function. Does citral have better anti-mildew property after being applied to bamboo? It is unknown yet. Therefore, only when the above problems are fully studied and discussed, can the promotion and application of citral in the field of bamboo mildew prevention be realized. Thus, for the first time to our knowledge, the authors studied the mechanism and antibacterial properties of citral against bamboo common mould, and explored the anti-mildew effect of bamboo treated with citral. This research will lay the foundation for the vigorous promotion and application of citral in the field of bamboo mildew prevention, and provide new ideas and open up new ways for broadening the application fields of natural antibacterial agents.

# 2. Material and methods

## 2.1. Materials

Citral (97%), sodium dihydrogen phosphate and disodium hydrogen phosphate were purchased from Sinopharm Chemical Reagent Co., Ltd (Shanghai, China). Tween-80 was obtained from Shanghai Lingfeng Chemical Reagent Co., Ltd (Shanghai, China). Moso bamboo strips (length 50 mm × width 20 mm × thickness 5 mm), with no knots and a moisture content of approximately 10%, was commercially obtained from Zhenghe Xizhuyuan Bamboo Products Factory. (Fujian, China). Test strains: PC, TV, AN and an equal mix of PC, TV and AN (Hun).

## 2.2. The preparation of mould suspensions

The jars containing an appropriate amount of sterile water and small glass beads were placed in a high-pressure steam sterilization pot and sterilized for 30 min at 121°C and 0.1 MPa. Afterwards, the mycelium and spores of the tested strains were picked with an inoculation needle under aseptic

condition and placed in sterilized jars. Finally, the mould suspensions were made by shaking for 10–15 min for inoculations.

## 2.3. The preparation of citral solution

A certain amount of citral was placed in a beater, then 2% (v/v) Tween-80, and a small amount of deionized water was added and mixed consistently. Next, they were moved into a volumetric flask for a constant volume. Finally, citral solutions with concentrations of 3.125, 6.25, 12.5, 25, 50 and 100 mg ml$^{-1}$ were obtained.

## 2.4. Oxford cup method to test the bacteriostatic performance of citral on bamboo mould

The Oxford cup method [19] was used to investigate the antibacterial property of citral on bamboo mould. First, the sterilized Oxford cups (8 mm outer diameter, 6 mm inner diameter) were placed on potato dextrose agar (PDA) plates with the size of 100 mm in diameter coated with 80 µl of the bacterial solution, then 80 µl of different concentrations of citral solution were injected into the cups. Second, the Petri dishes were sealed with sterile sealing film and pre-diffused for 2 h at 4°C. Finally, the Petri dishes were incubated at $28 \pm 2°C$ and relative humidity of $85 \pm 5\%$ for 2 days in the incubation chamber. The cross-crossing method was used to measure the diameter of the inhibition zone. Each treatment was repeated three times, the results were averaged and the bacteriostatic rates were calculated according to formula (2.1). Tween-80 treatment groups were used as controls:

$$\text{bacteriostatic rate} = \frac{\text{diameter of inhibition zone in treatment group} - \text{diameter of inhibition zone in control group}}{\text{diameter of inhibition zone in control group}} \times 100\%.$$

(2.1)

## 2.5. Minimal inhibitory concentration and minimal bactericidal concentration determinations

On the basis of the bacteriostatic results of citral, the concentration of citral was gradually diluted from 100 to 0 mg ml$^{-1}$, and the double dilution method [20] was used to test the minimal inhibitory concentration (MIC) and minimal bactericidal concentration (MBC) of citral. The PDA plates, which are 90 mm in diameter, and 15 times the inner diameter of the Oxford cup, containing different concentrations of citral were prepared by mixing different concentrations of citral uniformly, and then 80 µl of bacterial suspension was evenly coated on the surface of the PDA plates. Afterwards, the culture plates were sealed with a sterile sealing film and placed in a constant temperature incubator at $28 \pm 2°C$ and humidity of $85 \pm 5\%$. The growths of mould were observed after 2 days of culture, and the minimum concentration of citral for completely sterile growth was the MIC. Based on MIC determination, the culture was continued for 7 days and the MBC of completely sterile growth was taken as the MBC. Tween-80 treatment groups were the control ones.

## 2.6. Effects of citral on mycelia morphology of bamboo mould

A scanning electron microscope (SEM) was used to observe the effect of citral on mycelia morphology of bamboo mould. The mildew cakes with a diameter of 8.0 mm were laid on the flourishing culture medium for 7 days and placed on the surface of the medium plates with citral at various concentrations (0, MIC and MBC). Then, the samples were cultured at $28 \pm 2°C$ and $85 \pm 5\%$ humidity. After 4 days, the cakes were cut as the test samples under an electron microscope.

The samples were fixed with 2.5% glutaraldehyde at 4°C for 12 h. After that, the samples were washed three times with phosphate buffer solution (PBS, pH 7.0) for 15 min each. Subsequently, the samples were fixed with 1% osmium solution for 2 h. After fixing, the samples were rinsed three times with PBS solution for 15 min each. The samples were then dehydrated in ethanol series gradients (30, 50, 70, 80, 90 and 95%, v/v) for 15 min at a time. Finally, they were dehydrated with anhydrous ethanol for 20 min. The final samples were freeze-dried, gold-sprayed and observed in a SU8010 SEM.

## 2.7. Effects of citral on the microstructure of bamboo mould cells

Transmission electron microscopy (TEM) was used to observe the microstructures of mould cells. The preliminary treatment method of mould and electron microscope samples treated with citral was the same as that of §2.5, except that the samples were treated with pure acetone solution for 20 min after gradient dehydration with ethanol series. Then the samples were permeated, embedded, sliced and dyed. Finally, the microstructures of mycotic cells were observed in a JEM-1200 TEM.

## 2.8. Effects of citral on the release of cellular materials of bamboo mould

By simplifying Paul's method [21], the effects of citral on the release of cellular materials were investigated. Mould spores were cultured for 7 days, washed three times with PBS solution (pH 7.0) and suspended in buffer solution. Then appropriate spore suspensions were taken and treated with citral with concentrations of 0, MIC, MBC for 0, 30, 60 and 120 min, respectively. Afterward, 5 ml samples were collected and centrifuged at 12 000 r.p.m. for 5 min. Then, the supernatant was taken and the absorbance measured at 260 nm with a UV-1800 spectrophotometer. The control group was calibrated with PBS (pH 7.0).

## 2.9. Effect of citral on the extracellular pH of bamboo mould cells

The extracellular pH of mould cells treated with citral was measured with a micro pH/mV metre. The treatment method of spore suspensions was identical to that in §2.8. After treatment, 5 ml spore suspensions were taken to measure the extracellular pH, and each group was repeated three times. Tween-80 treatment groups were the control ones.

## 2.10. Anti-mildew test of bamboo treated with citral solution

Bamboo strips were placed in a pressurized tank with citral solution, and then removed to dry for later use. Then, anti-mould properties of citral were examined according to the 'Test method for anti-mildew agents in controlling wood mould and stain fungi' (GB/T 18261-2013) [22]: spore suspensions of PC, TV, AN, and Hun were smeared in Petri dishes containing the plate medium. After 2 min, a sterilized U-shaped solid glass rod was put on every Petri dish. Next, the Petri dishes were placed in an incubator at a temperature of $28 \pm 2°C$ and a relative humidity of $85 \pm 5\%$ for mildew cultivation. After the moulds were successfully cultivated, the bamboo strips were placed on the U-shaped glass rods, and the edge of the Petri dishes were sealed with parafilm; this was repeated three times for each group. Finally, the Petri dishes with bamboo strips were placed into the incubator for a mildew resistance test. Every other day, the bamboo strips in the incubator infected by PC, TV, AN and Hun were observed and recorded and the infection values took the average of the results (table 3). On day 28, photographs of the bamboo samples were taken, and the area of the bamboo strips infected by the fungus were observed and analysed to determine the infection levels of the bamboo strips (table 1) and the prevention and control effectiveness were calculated according to formula (2.2), and the anti-mould properties of the citral were analysed:

$$E = \left(1 - \frac{D_1}{D_0}\right) \times 100\%, \tag{2.2}$$

where $E$ is the anti-mould efficiency (%), $D_1$ is the average infection ratio of extracted, specimens and $D_0$ is the average infection ratio of control specimens. The anti-mould, efficiency of a specific group of specimens is defined as the mean value of their $E$ values, against the three individual mildews and the mildew mix.

# 3. Results

## 3.1. Inhibitory property

The Oxford cup method was used to study the effect of different concentrations of citral on the inhibitory zone diameter of PC, TV, AN and Hun, and the inhibitory zone diameters were calculated to characterize the anti-mould ability of citral. The results are shown in figures 1 and 2, and table 2.

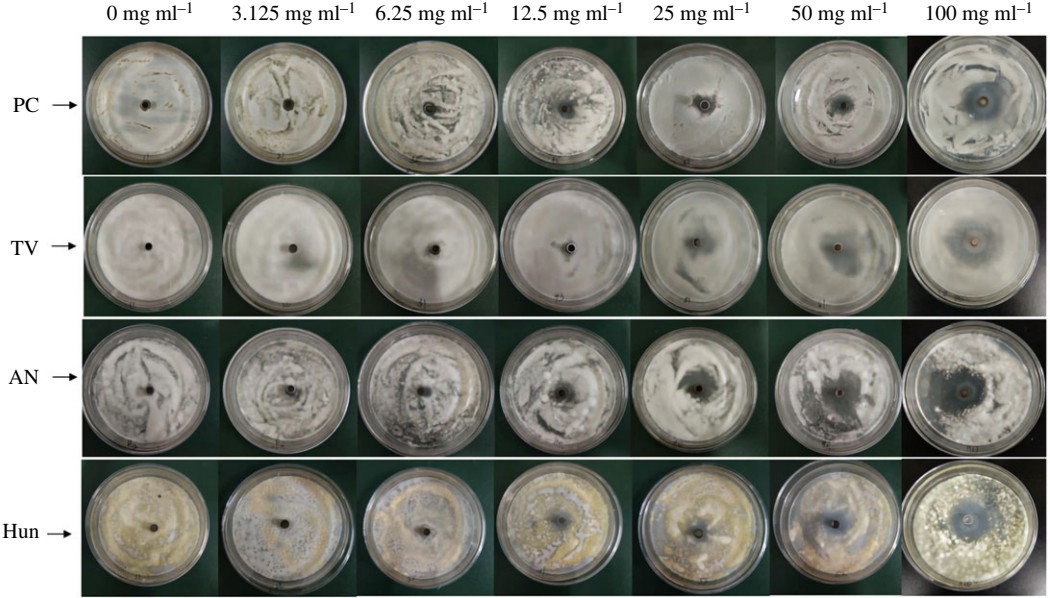

**Figure 1.** Inhibitory effect of citral against bamboo mould (PC, TV, AN, Hun were *Penicillium citrinum*, *Trichoderma viride*, *Aspergillus niger* and a hybrid fungi group comprising PC, TV and AN, respectively).

**Table 1.** Classification standard of surface infection levels of samples.

| infection value | infected area of sample |
|---|---|
| 0 | no hyphae or mildew on the sample surface |
| 1 | infected area of sample <1/4 |
| 2 | infected area of sample 1/4–1/2 |
| 3 | infected area of sample 1/2–3/4 |
| 4 | infected area of sample >3/4 |

Citral has inhibitory zones on PC, TV, AN and Hun (figure 1), indicating that citral has an inhibitory effect on common bamboo mould. With an increase in citral concentration, the diameters of the inhibitory zone and antibacterial rates of PC, TV, AN and Hun increased (figure 2 and table 2). When the concentration of citral was 100 mg ml$^{-1}$, the diameters of the inhibitory zone of PC, TV, AN and Hun were 31.57, 32.08, 36.64 and 30.42 mm, respectively, which were 3.00, 4.58, 4.92 and 4.38 fold of the control group. The bacteriostatic rates were 200.95, 358.29, 392.47 and 338.32%, respectively. It can be seen that the higher the concentration of citral, the better the bacteriostatic performance. Because 100 mg ml$^{-1}$ citral has an inhibitory rate of over 100% against PC, TV, AN and Hun, citral has good antibacterial properties against these four types of moulds, among which the bacteriostatic performance on AN was the best, and that of PC was relatively poor.

## 3.2. Minimal inhibitory concentration and minimal bactericidal concentration

On the basis of the bacteriostatic results of citral, the concentration of citral was gradually diluted from 100 to 0 mg ml$^{-1}$, and the double dilution method was used to test the MIC and MBC of citral to PC, TV, AN and Hun. The results are shown in table 3.

The MIC of citral against PC, TV, AN and Hun was 0.180, 0.265, 0.226 and 0.233 mg ml$^{-1}$, respectively (table 3). The MIC of the four moulds was very low, among which the MIC of PC was the lowest. This indicated that citral had a greater inhibitory effect on PC growth of the four moulds and could inhibit its growth at a low concentration. To inhibit the growth of the three moulds simultaneously, the MIC of Hun should theoretically be no less than that of a single mould. However, the results showed that the MIC of

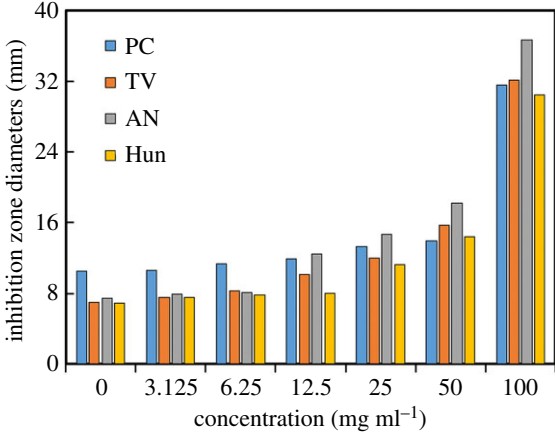

**Figure 2.** Effects of citral on the diameters of the inhibition zones of bamboo mould (PC, TV, AN, Hun are *Penicillium citrinum*, *Trichoderma viride*, *Aspergillus niger* and a hybrid fungi group comprising PC, TV and AN, respectively).

**Table 2.** Effects of citral on the inhibition rates of bamboo mould (PC, TV, AN, Hun are *Penicillium citrinum*, *Trichoderma viride*, *Aspergillus niger* and a hybrid fungi group comprising PC, TV and AN, respectively).

| concentration (mg ml$^{-1}$) | inhibition rates (%) | | | |
|---|---|---|---|---|
| | PC | TV | AN | Hun |
| 3.125 | 1.13 | 8.29 | 6.32 | 8.79 |
| 6.250 | 8.20 | 18.00 | 8.33 | 13.11 |
| 12.500 | 13.82 | 44.86 | 67.61 | 15.13 |
| 25.000 | 26.88 | 71.14 | 97.04 | 61.82 |
| 50.000 | 32.60 | 124.57 | 144.89 | 107.20 |
| 100.000 | 200.95 | 358.29 | 392.47 | 338.32 |

**Table 3.** Minimal inhibitory concentration (MIC) and minimal bactericidal concentration (MBC) of citral on bamboo mould (PC, TV, AN, Hun are *Penicillium citrinum*, *Trichoderma viride*, *Aspergillus niger* and a hybrid fungi group comprising PC, TV and AN, respectively).

| mould name | MIC (mg ml$^{-1}$) | MBC (mg ml$^{-1}$) |
|---|---|---|
| PC | 0.180 | 0.499 |
| TV | 0.265 | 0.495 |
| AN | 0.226 | 0.381 |
| Hun | 0.233 | 0.509 |

mixed mould was more than that of PC and AN, and smaller than that of TV. This indicated that under a certain living environment, there is a competitive relationship between the three moulds, so the concentration of citral does not need to reach the maximum MIC of a single mould to inhibit the three moulds simultaneously. The MBC of citral against PC, TV, AN and Hun was 0.499, 0.495, 0.381 and 0.509 mg ml$^{-1}$, respectively (table 3), among which the MBC of AN was the lowest. This indicated that different types of moulds have different tolerance levels to citral, among which AN has the least tolerance to citral. The MBC of PC, TV, AN and Hun were higher than MIC, which was 2.77, 1.87, 1.67 and 2.18 fold, respectively. Therefore, to completely kill the mould, the treatment concentration of citral must be increased.

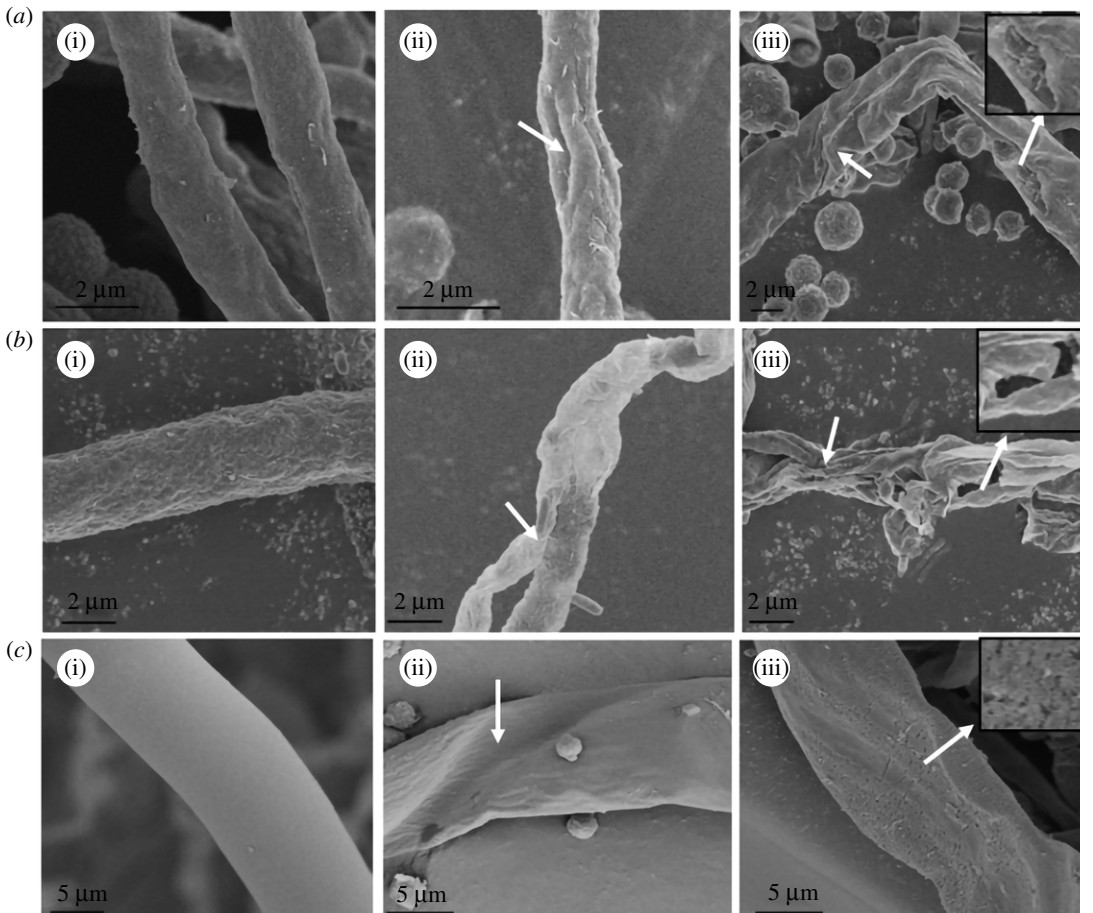

**Figure 3.** Effect of citral on mycelia morphology of bamboo mould (*a*(i), *a*(ii) and *a*(iii) are the PC control group, minimal inhibitory concentration treatment group and minimal bactericidal concentration treatment group, respectively; *b*(i), *b*(ii) and *b*(iii) are the TV control group, minimal inhibitory concentration treatment group and minimal bactericidal concentration treatment group, respectively; *c*(i), *c*(ii) and *c*(iii) are the AN control group, minimal inhibitory concentration treatment group and minimal bactericidal concentration treatment group, respectively).

## 3.3. Effects of citral on mycelia morphology of bamboo mould

MIC and MBC of citral were used to treat bamboo mould, and then the effects of citral on the mycelia morphology of bamboo mould were observed by SEM. The results are shown in figure 3.

In the control group (figure 3*a*(i), *b*(i) and *c*(i)), the mycelia surface of mould was relatively smooth, with full and regular shape, uniform thickness and complete structure. In the MIC group, the mycelium of PC (figure 3*a*(ii)) appeared dry and of uneven surface and thickness. TV treatment (figure 3*b*(ii)) showed two mycelia, which were severely shrivelled and distorted. The mycelium of AN (figure 3*c*(ii)) was withered, the surface was wrinkled and the whole was irregular and distorted. In the MBC treatment group, the mycelium of PC (figure 3*a*(iii)) was not only dry and rough but also appeared partially ruptured. The mycelium of the TV (figure 3*b*(iii)) was severely shrivelled, with a distorted shape and broken structure. The mycelium of AN (figure 3*c*(iii)) was severely shrivelled, its shape was seriously distorted, its surface was rough, and a large number of holes appeared, and its structure became loose. It can be seen that different concentrations of citral damaged mycelium shape and structural integrity. The greater the concentration of citral is, the greater the degree of damage would be. Therefore, it can be inferred that citral can make the mycelium severely shrivelled, distorted, rough, and perforated or ruptured to inhibit or kill the mould.

## 3.4. Effects of citral on cellular structures of bamboo mould

Figure 4 shows the cellular structures of PC, TV, AN and Hun treated with MIC and MBC of citral.

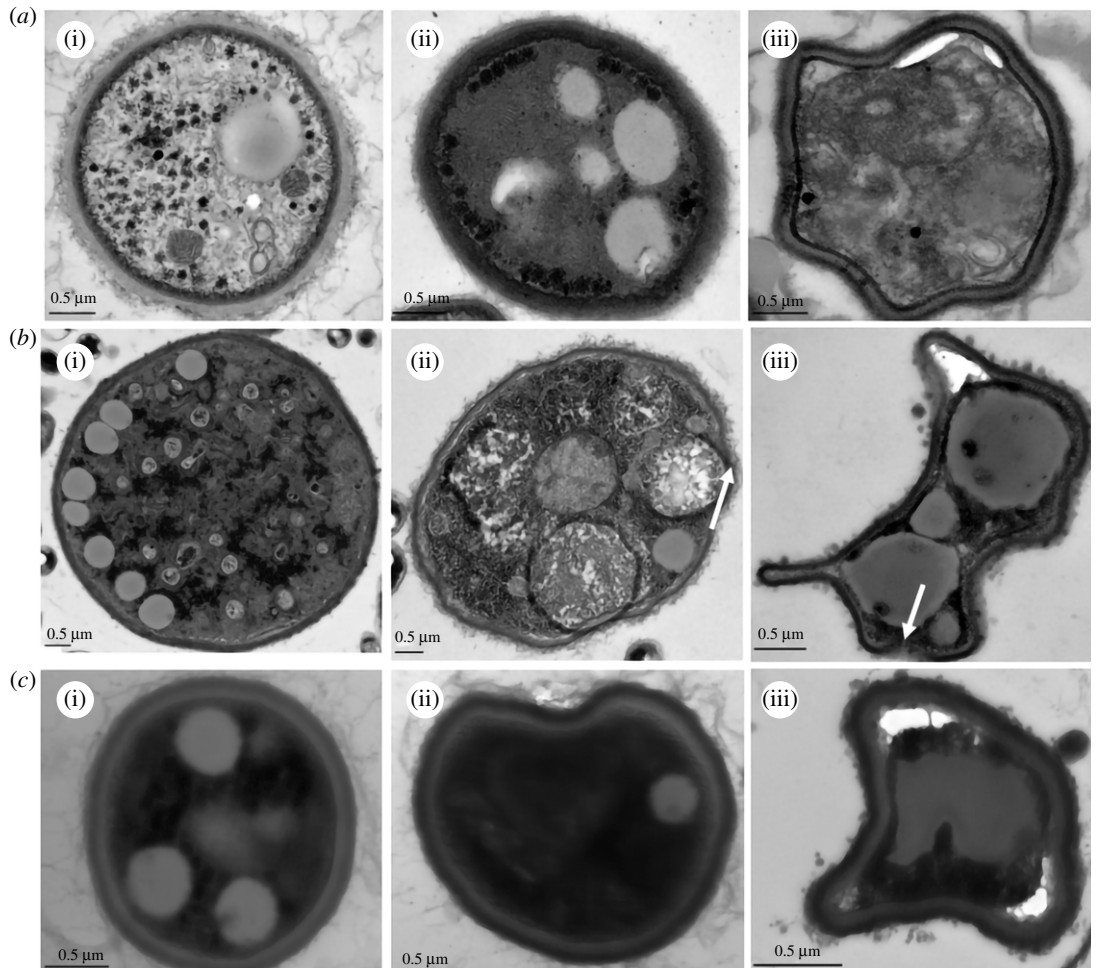

**Figure 4.** Effects of citral on cellular structures of bamboo mould (*a*(i), *a*(ii) and *a*(iii) are the PC control group, minimal inhibitory concentration treatment group and minimal bactericidal concentration treatment group, respectively; *b*(i), *b*(ii) and *b*(iii) are the TV control group, minimal inhibitory concentration treatment group and minimal bactericidal concentration treatment group respectively; *c*(i), *c*(ii) and *c*(iii) are the AN control group, minimal inhibitory concentration treatment group and minimal bactericidal concentration treatment group, respectively).

In the control group (figure 4*a*(i), *b*(i) and *c*(i)), mycelial cell structures were intact, organelles were uniformly distributed in the cytoplasm, protoplasts were uniformly dense, cell wall thicknesses were consistent and cells were in a normal growth state. In the MIC group, for PC (figure 4*a*(ii)), the membrane-free transparent inclusions increased, cell wall blurred and the intracellular colour deepened. For TV (figure 4*b*(ii)), the cell was slightly deformed, with uneven distribution of intracellular materials, large vacuoles and a large number of internal vesicles formed, which accumulated into large cavities and dissolved out of the cell. The mycelium cell wall of AN (figure 4*c*(ii)) collapsed, and the cell was deformed. In the MBC treatment group, the cell of PC (figure 4*a*(iii)) was invaginated and deformed, with cytoplasmic wall separation and extravasation. The cell wall of TV (figure 4*b*(iii)) was thinner than that of AN and PC, which resulted in the lysis of the cell wall and cell membrane, severe deformation, plasmodium separation, serious aggregation of contents cytoplasmic loss, etc. The cytoplasmic wall of AN (figure 4*c*(iii)) was isolated, and a dark staining substance appeared outside the cell. Different concentrations of citral damage the mould cell structure. The higher concentration of citral resulted in a greater degree of damage to the cell structure. As AN, TV and PC belong to multi-cellular fungi, the cell wall of the fungus maintains its inherent shape, inhibits mechanical and osmotic damage, and acts as a barrier. The cell membrane plays a crucial role in maintaining cell balance, material exchange and energy transfer. Once the integrity of the cell membrane is destroyed, a series of reactions, such as changes in membrane permeability and dissolution of intracellular substances, will accelerate the inactivation of the cell [23]. Therefore, the damage of citral to mould is mainly through changing the cell shape, destroying the

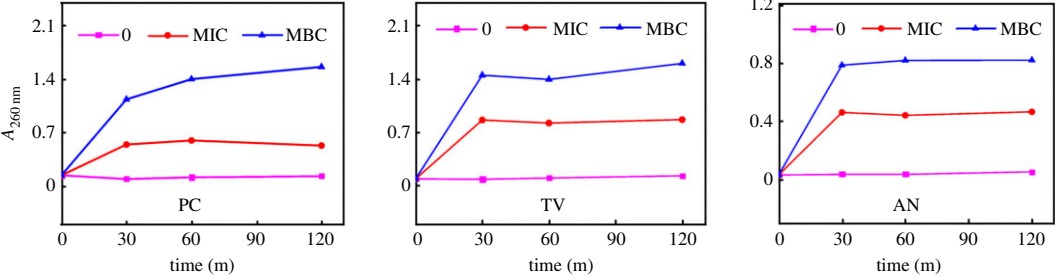

**Figure 5.** Effects of citral on absorbance at 260 nm of extracellular fluids of bamboo mould (0: control group; MIC: minimal inhibitory concentration treated group; MBC: minimal bactericidal concentration treated group; PC, TV, AN, Hun are *Penicillium citrinum*, *Trichoderma viride*, *Aspergillus niger* and a hybrid fungi group comprising PC, TV and AN, respectively).

integrity of the cell wall and membrane structure, resulting in loss of cytoplasm; material aggregation and distribution in the cell is not uniform, to make mould autolysis or inhibit its life activities.

## 3.5. Effects of citral on the release of cellular components of the bamboo mould

The absorbance of the extracellular fluid was measured at 260 nm after different concentrations of citral were treated for different time intervals. The results are shown in figure 5.

Under normal circumstances, the absorbance of mould at 260 nm increases significantly, which means a large amount of nucleic acid, protein and other substances leak from the cell, causing irreversible damage to the cell membrane and cytoplasm [24]. The absorbance of PC, TV, AN and the control group was unchanged within the range of 0–120 min, and the curve was approximately a straight line (figure 5), which indicated that the nucleic acid, protein and other substances in the extracellular fluid of mould in the control group were very few, and mould cells in the control group were not destroyed. In both the MIC treatment group for PC, TV and AN and the MBC treatment group, the treatment time of 30 min was a turning point. Within the range of 0–30 min, the absorbance increased sharply with an increase in treatment time. After 30 min and within 30–120 min with an increase in processing time, absorbance increased or scarcely increased and the curves were flattened. This showed that after bamboo mould was treated with citral, leakage of nucleic acids and proteins in the extracellular fluid increased with the increase in processing time, the destruction of the mould level also increased. However, after 30 min of treatment, the increase tended to be stable. Figure 5 also shows that the absorbances of the three moulds of the MBC treatment group were higher than that of the MIC treatment group. Therefore, the higher the concentration of citral in mould treatment, the longer the treatment time, the greater the degree of damage to mould cells, resulting in more leakage of nucleic acid, protein and other substances in the cell. The greater the absorbance, which is consistent with the result that the higher the concentration of citral in mould cells, the greater the damage of hypha morphology and cell structure of mould cells.

## 3.6. Effects of citral on extracellular pH of bamboo mould

The pH of the extracellular pH of bamboo mould treated with different concentrations of citral for different periods was detected by a micro pH/mV metre, and the results are shown in figure 6.

pH is a key factor that controls DNA transcription, protein synthesis, enzyme activity and so on. In order to maintain pH stability, a dynamic balance exists between acid and alkali production in cells. In normal cells, the extracellular pH is neutral, while the extracellular pH is slightly acidic. Therefore, when extracellular fluid is slightly acidic, the secretion of protons inside the cell will be reduced to balance the intracellular and extracellular fluids. The extracellular pH decreased as a whole during 0–30 min of treatment but decreased more slowly in the treatment group than in the control group, indicating that during this treatment period cells would secrete acid to reduce the pH of extracellular fluid, but cells in the treatment group would suffer certain damage, leading to the outflow of a small number of protons and the pH would decrease more slowly. After 30–60 min of treatment, the pH of the extracellular fluid in the control group and the treatment group increased as a whole because when the pH of the extracellular fluid was lower than that of intracellular fluid, the cells would self-regulate and begin to produce acid, reduce or not secret acid; thus, increasing extracellular pH. After 60 min of

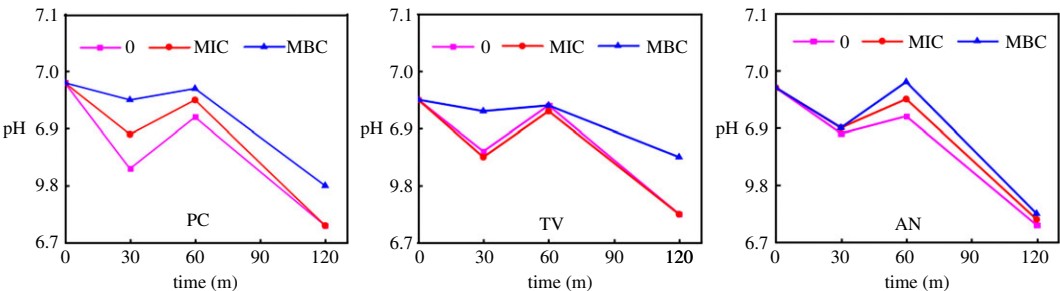

**Figure 6.** Effect of citral on pH value of extracellular fluid of bamboo mould (0: control group; MIC: minimal inhibitory concentration treated group; MBC: minimal bactericidal concentration treated group; PC, TV, AN, Hun are *Penicillium citrinum*, *Trichoderma viride*, *Aspergillus niger* and a hybrid fungi group comprising PC, TV and AN, respectively).

treatment, the extracellular pH began to decrease gradually owing to the large amount of acid produced in the cell secreted into the extracellular environment when the mould was subjected to external action for a long time, resulting in a decrease in extracellular pH. The extracellular pH of bamboo mould presented a downward trend with the extension of processing time. The pH values of the MBC treatment group were higher than that of the control group in the same period (figure 6). This indicated that the control group also had a certain influence on the pH value of the extracellular fluid. However, at the same time, the higher the concentration of citral, the higher the extracellular pH of mould. It can be inferred that citral can interfere with or destroy the pH balance of intracellular and extracellular fluid to achieve the inhibition or killing effect of mould.

## 3.7. Anti-mildew test of bamboo treated with citral

When the concentration of citral was 100 mg ml$^{-1}$, the inhibition rate of all the four kinds of moulds reached more than 100%. Therefore, citral with concentrations of 100, 150 and 200 mg ml$^{-1}$ were used to treated bamboo strips to prevent mildew, and the anti-mould effects of the treated bamboo strips and the untreated bamboo strips (control group) were tested for 28 days. The results are shown in figure 7 and table 3.

Figure 7 shows that the surface of the bamboo strips in the control group was covered with PC, TV, AN and Hun hyphae, respectively, after 28 days, indicating that the control group had no control effect on bamboo mould. In the 100 mg ml$^{-1}$ citral treatment group, the surface of the bamboo strips was were covered by a large amount of PC, AN and Hun, respectively, and the growth of mycelium was not significantly different from that of the control group, but there was only a small amount of TV growth on the surface of the treated bamboo strips, indicating that the 100 mg ml$^{-1}$ citral treatment of the bamboo has no effect on PC, AN and Hun and has a certain anti-mildew effect on TV. In the 150 mg ml$^{-1}$ citral treatment group, only a small amount of PC, AN and Hun grew on the surface of the treated bamboo strips, but TV does not grow, indicating that 150 mg ml$^{-1}$ citral has a certain control effect on the four kinds of moulds. In the 200 mg ml$^{-1}$ citral treatment group, no moulds grow on the surface of the bamboo strips, indicating that it has a good control effect on the four kinds of moulds. Table 4 shows that only 200 mg ml$^{-1}$ of the citral treatment group of bamboo strips has 100% control effect on the four kinds of moulds. This shows that the mildew proof effect can be achieved only if the concentration of citral to treat bamboo is increased to more than twice the concentration of citral directly acting on mould. This may be because citral is easy to volatilize, and the processed bamboo strips need to be dried before use, which leads to a large loss of citral. Therefore, when citral is used as an anti-mould agent for bamboo, in order to obtain a better anti-mildew effect, the concentration of citral must be doubled.

## 4. Conclusion

With an increase in citral concentration, the antibacterial rates of PC, TV, AN and Hun increased. When the concentration of citral is 100 mg ml$^{-1}$, the antibacterial rates of citral against PC, TV, AN and Hun were more than 100%, among which the bacteriostatic performance on AN was the best, and that of PC was relatively poor.

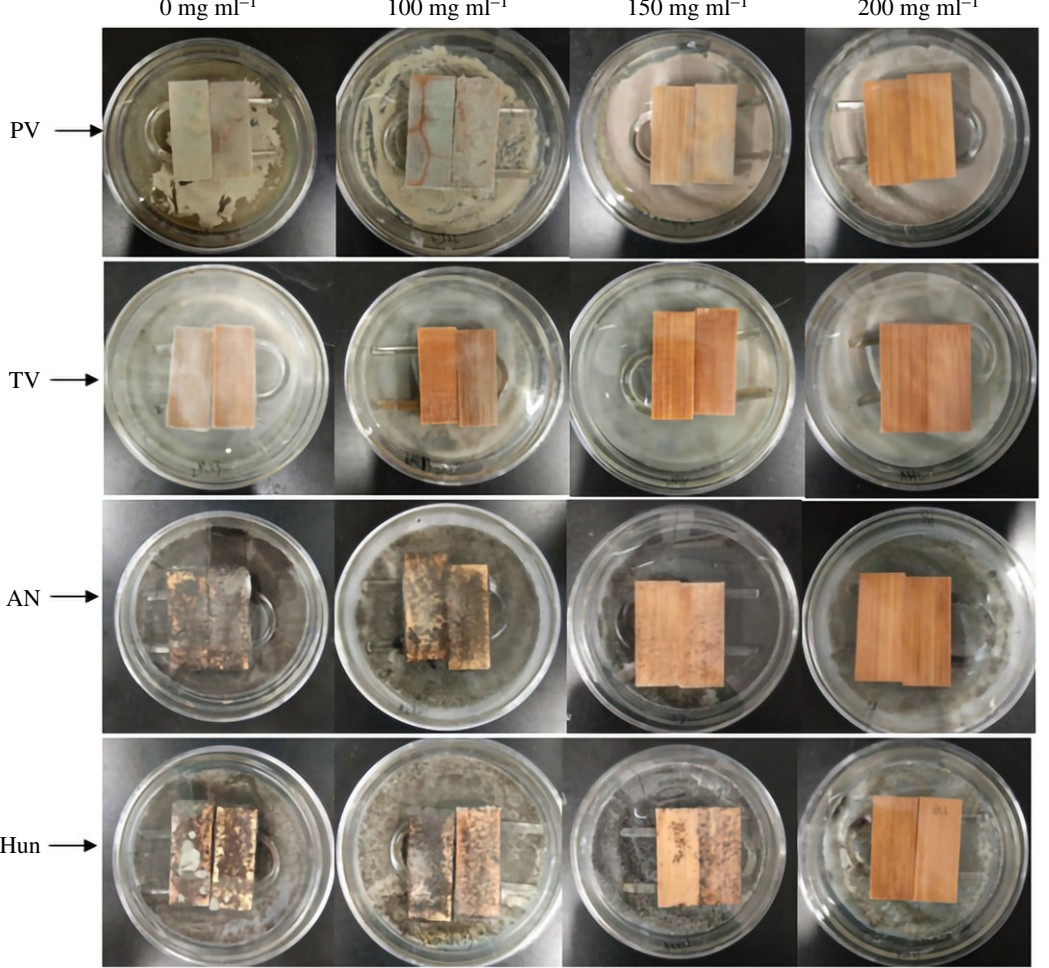

**Figure 7.** The anti-mildew photographs of bamboo strips treated with citral at different concentrations on the 28th day (PC, TV, AN, Hun are *Penicillium citrinum*, *Trichoderma viride*, *Aspergillus niger* and a hybrid fungi group comprising PC, TV and AN, respectively).

**Table 4.** Anti-mould efficiency of treated bamboo strips on the 28th day.

| concentration (mg ml$^{-1}$) | anti-mould efficiency (%) | | | |
|---|---|---|---|---|
| | PC | TV | AN | Hun |
| 100 | 0 | 83.25 | 0 | 0 |
| 150 | 33.25 | 100 | 66.75 | 16.75 |
| 200 | 100 | 100 | 100 | 100 |

The MIC of citral against PC, TV, AN and Hun was 0.180, 0.265, 0.226 and 0.233 mg ml$^{-1}$, respectively. The MBC of citral against PC, TV, AN and Hun was 0.499, 0.495, 0.381 and 0.509 mg ml$^{-1}$, respectively. The MBC of citral against PC, TV, AN and Hun was higher than MIC, which was 2.77, 1.87, 1.67 and 2.18 fold, respectively. Therefore, increasing the concentration of citral can effectively kill mould.

Different concentrations of citral can damage the mycelium shape, structural integrity, cell structure, etc. The higher the concentration of citral and the longer the treatment time, the greater the degree of damage to mould cells.

Citral can change the shape of mycelium and the cell, destroy the integrity of mycelium structure, cell wall and membrane structure, which in turn causes the loss of cytoplasm and aggregation and uneven

distribution of substances in the cell, leakage of nucleic acid, protein and other substances in the cell, as well as destroying the pH balance inside and outside of the cell.

Compared with the direct effect of citral on mould, the antibacterial property of bamboo treated with citral was significantly reduced, the mildew proof effect can be achieved only if the concentration of citral to treat bamboo is increased to more than twice the concentration of citral directly acting on mould.

Data accessibility. Data is available from the Dryad Digital Repository: http://dx.doi.org/10.5061/dryad.c2fqz616z [25].
Authors' contributions. J.Z. carried out the work, participated in data analysis, participated in the design of the study and drafted the manuscript; C.D. conceived of the study, designed the study and critically revised the manuscript; Q.L., A.H. and R.P. helped to perform the experiments and analyse part data; F.S. and W.Z. provided experimental guidance. All authors gave final approval for publication and agree to be held accountable for the work performed therein.
Competing interests. We declare we have no competing interests.
Funding. This work was supported by the National Natural Science Foundation of China (grant no. 31870541), Zhejiang Provincial Key Research and Development Project (grant no. 2019C02037) and National Key Research and Development Plan of the '13th Five-Year' of China (grant no. 2017YFD0601105).
Acknowledgements. We thank Ms Hui Wang for support in the antibacterial test and mould-resistant test experimental work.

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
