## [Peer Review File · Royal Society Open Science]

Review History

RSOS-202244.R0 (Original submission)

Review form: Reviewer 1

Is the manuscript scientifically sound in its present form?

Yes

Are the interpretations and conclusions justified by the results?

Yes

Is the language acceptable?

Yes

Do you have any ethical concerns with this paper?

No

Have you any concerns about statistical analyses in this paper?

No

Recommendation?

Accept with minor revision (please list in comments)

Comments to the Author(s)

Reviewer comments (RSOS-202244):

The authors investigated inhibition mechanism and antibacterial activity of natural antibacterial agent citral on bamboo mold and its anti- mildew effect on bamboo. The novelty of the article is very high. The study is the pioneering research on the application of citral to the mildew prevention of bamboo. Before publication, authors are required to revise the manuscript with minor modifications as under:

1. Page 5, Line 29. The sentence "When the concentration of citral was 100 mg/ml, the diameters of the inhibitory zone of PC, TV, AN, and Hun were 31.57、 32.08、 36.64mm and 30.42mm, respectively" should be replaced by "The sentence "When the concentration of citral was 100 mg/ml, the diameters of the inhibitory zone of PC, TV, AN, and Hun were 31.57, 32.08, 36.64, and 30.42 mm, respectively".
2. The unit "mg/mL" should be unified in the manuscript. "mg/mL" or "mg/ml".
3. Table 2 and 3 are hard to understand. Please revise it.
4. Figure 5 and 6 are not clearly to see, please adjust font size in the figures.

Review form: Reviewer 2

Is the manuscript scientifically sound in its present form?

No

Are the interpretations and conclusions justified by the results?

Yes

Is the language acceptable?

Yes

Do you have any ethical concerns with this paper?

No

Have you any concerns about statistical analyses in this paper?

No

Recommendation?

Major revision is needed (please make suggestions in comments)

Comments to the Author(s)

My main issue with this work is why bamboo? Why is this different for bamboo than other substrates such as wood? Bamboo is fascinating to the scientific community due to its graded microstructure (lit review is weak here too). Is that fact play a role? Why is the graded structure of bamboo makes a difference? Is there a role in bamboo cell wall sizes?

the presentation needs to significantly improve. Figures are missing explanations of the abbreviation and figure captions are short.

Decision letter (RSOS-202244.R0)

Dear Ms Zhang

The Editors assigned to your paper RSOS-202244 "Inhibition mechanism and antibacterial activity of natural antibacterial agent citral on bamboo mold and its anti- mildew effect on bamboo" have now received comments from reviewers and would like you to revise the paper in accordance with the reviewer comments and any comments from the Editors. Please note this decision does not guarantee eventual acceptance.

Please submit your revised manuscript and required files (see below) no later than 21 days from today's (ie 11-Mar-2021) date. Note: the ScholarOne system will 'lock' if submission of the revision is attempted 21 or more days after the deadline. If you do not think you will be able to meet this deadline please contact the editorial office immediately.

on behalf of Professor Luning Liu (Associate Editor) and Catrin Pritchard (Subject Editor)
openscience@royalsociety.org

Associate Editor Comments to Author (Professor Luning Liu):

Please revise the manuscript thoroughly based on the comments of Reviewer 1 and Reviewer 2.

Reviewer comments to Author:

Reviewer: 1

Comments to the Author(s)

Reviewer comments (RSOS-202244):

The authors investigated inhibition mechanism and antibacterial activity of natural antibacterial agent citral on bamboo mold and its anti- mildew effect on bamboo. The novelty of the article is very high. The study is the pioneering research on the application of citral to the mildew prevention of bamboo. Before publication, authors are required to revise the manuscript with minor modifications as under:

1. Page 5, Line 29. The sentence "When the concentration of citral was 100 mg/ml, the diameters of the inhibitory zone of PC, TV, AN, and Hun were 31.57、 32.08、 36.64mm and 30.42mm, respectively" should be replaced by "The sentence "When the concentration of citral was 100 mg/ml, the diameters of the inhibitory zone of PC, TV, AN, and Hun were 31.57, 32.08, 36.64, and 30.42 mm, respectively".
2. The unit "mg/mL" should be unified in the manuscript. "mg/mL" or "mg/ml".
3. Table 2 and 3 are hard to understand. Please revise it.
4. Figure 5 and 6 are not clearly to see, please adjust font size in the figures.

Reviewer: 2

Comments to the Author(s)

My main issue with this work is why bamboo? Why is this different for bamboo than other substrates such as wood? Bamboo is fascinating to the scientific community due to its graded microstructure (lit review is weak here too). Is that fact play a role? Why is the graded structure of bamboo makes a difference? Is there a role in bamboo cell wall sizes?

the presentation needs to significantly improve. Figures are missing explanations of the abbreviation and figure captions are short.

===PREPARING YOUR MANUSCRIPT===

If you have been asked to revise the written English in your submission as a condition of publication, you must do so, and you are expected to provide evidence that you have received

language editing support. The journal would prefer that you use a professional language editing service and provide a certificate of editing, but a signed letter from a colleague who is a native speaker of English is acceptable. Note the journal has arranged a number of discounts for authors using professional language editing services (<https://royalsociety.org/journals/authors/benefits/language-editing/>).

===PREPARING YOUR REVISION IN SCHOLARONE===

<https://royalsociety.org/journals/authors/author-guidelines/#supplementary-material> to

include a suitable title and informative caption. An example of appropriate titling and captioning may be found at https://figshare.com/articles/Table_S2_from_Is_there_a_trade-off_between_peak_performance_and_performance_breadth_across_temperatures_for_aerobic_sc_ope_in_teleost_fishes_/3843624.

Author's Response to Decision Letter for (RSOS-202244.R0)

See Appendix A.

Decision letter (RSOS-202244.R1)

Dear Ms Zhang,

It is a pleasure to accept your manuscript entitled "Inhibition mechanism and antibacterial activity of natural antibacterial agent citral on bamboo mold and its anti- mildew effect on bamboo" in its current form for publication in Royal Society Open Science.

Best regards,

Lianne Parkhouse
Editorial Coordinator
Royal Society Open Science

on behalf of Professor Luning Liu (Associate Editor) and Catrin Pritchard (Subject Editor)
openscience@royalsociety.org

Appendix A

Response to comments of reviewers and editors

Reviewer1:

The authors investigated inhibition mechanism and antibacterial activity of natural antibacterial agent citral on bamboo mold and its anti- mildew effect on bamboo. The novelty of the article is very high. The study is the pioneering research on the application of citral to the mildew prevention of bamboo. Before publication, authors are required to revise the manuscript with minor modifications as under:

1. Page 5, Line 29. The sentence “When the concentration of citral was 100 mg/ml, the diameters of the inhibitory zone of PC, TV, AN, and Hun were 31.57、 32.08、 36.64mm and 30.42mm, respectively” should be replaced by “The sentence “When the concentration of citral was 100 mg/ml, the diameters of the inhibitory zone of PC, TV, AN, and Hun were 31.57, 32.08, 36.64, and 30.42 mm, respectively”.

Response: Thank you for your advice. It has been modified as suggested and shown as the follow:

When the concentration of citral was 100 mg/ml, the diameters of the inhibitory zone of PC, TV, AN, and Hun were 31.57, 32.08, 36.64, and 30.42 mm, respectively

2. The unit “mg/mL” should be unified in the manuscript. “mg/mL” or “mg/ml”.

Response: Thank you for your suggestion. According to the suggestion, they have been unified in the manuscript. The concentration unit is “mg/ml”.

3. Table 2 and 3 are hard to understand. Please revise it.

Response: I'm really sorry. Maybe because of the typesetting of the manuscript, the tables were confused after the manuscript was uploaded, which made them difficult to understand. We have adjusted them and shown as follows:

Table 2. Effects of citral on the inhibition rates of bamboo mold.

Concentration (mg/ml)	Inhibition rates (%)			
	PC	TV	AN	Hun
3.125	1.13	8.29	6.32	8.79
6.250	8.20	18.00	8.33	13.11
12.500	13.82	44.86	67.61	15.13
25.000	26.88	71.14	97.04	61.82
50.000	32.60	124.57	144.89	107.20
100.000	200.95	358.29	392.47	338.32

Table 3. Minimal inhibitory concentration (MIC) and minimal bactericidal concentration (MBC) of citral on bamboo mold.

Mold name	MIC (mg/ml)	MBC (mg/ml)
PC	0.180	0.499
TV	0.265	0.495
AN	0.226	0.381
Hun	0.233	0.509

4. Figure 5 and 6 are not clearly to see, please adjust font size in the figures.

Response: Thank you for your suggestion. According to the reviewer’s comments, we have adjusted them, as follows:

Figure 5. Effects of citral on absorbance at 260nm of extracellular fluids of bamboo mold (0: control group; MIC: minimal inhibitory concentration treated group; MBC: minimal bactericidal concentration treated group).

Figure 6. Effect of citral on pH value of extracellular fluid of bamboo mold(0: control group; MIC: minimal inhibitory concentration treated group; MBC: minimal bactericidal concentration treated group).

Reviewer 2:

1. My main issue with this work is why bamboo?

Response: Thanks for your kind question. The increasing demand for the limited forest resources in various applications, has led to the shortage in wood supply. Thus, there is an urgent need to look for new materials as alternatives of wood. Bamboo is the fastest growing woody plants in the world, which grows to its maximum height in about 3 months and reaches maturity in 3~4 years. Due to its natural aesthetic beauty, incredible strength, and an advantage as a sustainable and eco-friendly substitute for increasingly depleted wood resources, bamboo has been taken as a notable economical and versatile raw material extensively used in the fields of decoration, architecture and furniture. However, bamboo is easy to mildew, and mildew causes the surface contamination of bamboo, showing brown or black, which causes bamboo to lose its value. Thus, it is imperative to prevent mildew in bamboo. In order to overcome this problem and enhance the value of bamboo, in this work, we explored the anti-mildew effect of bamboo treated with citral.

2. Why is this different for bamboo than other substrates such as wood?

Response: Thanks for your kind question. Compared with wood, bamboo has fundamental differences in growth, structure and property: bamboo has no growth rings, and its height growth is completed in 2~4 months, and there is no diameter growth after height growth, unlike trees, which grow in height and diameter for decades; The vascular bundles of bamboo are all arranged longitudinally without transverse ray tissue; Bamboo contains more sugar, starch, protein and so on than wood, which is more susceptible to mildew and decay.

3. Bamboo is fascinating to the scientific community due to its graded microstructure (lit review is weak here too). Is that fact play a role? Why is the graded structure of bamboo makes a difference? Is there a role in bamboo cell wall sizes?

Response: Thanks for your kind question. Bamboo can be seen as a two-phase composite material with multi-level structure composed of vascular bundles and basic tissues. The fiber sheath in the vascular bundle can be regarded as a reinforcing phase, with a large aspect ratio and a thicker cell wall. The basic structure is composed of parenchyma cells, which can be regarded as the matrix phase. The perfect combination of vascular bundles and basic tissues is the fundamental reason that bamboo has excellent mechanical properties, which makes the specific strength of bamboo 3~4 times higher than that of steel. Bamboo is also a typical natural gradient material. The distribution of vascular bundle of bamboo along the direction of bamboo wall thickness shows a gradient. The vascular bundle distribution is dense in the outer region and sparse in the inner region. The gradient structure of bamboo makes the permeability of bamboo much worse than that of wood, which leads to the effect of pressure impregnation treatment of bamboo worse than that of wood. Therefore, it is necessary to study the effect of pressure impregnation and mould-proof treatment of bamboo.

4.the presentation needs to significantly improve.

Response: Thank you for your suggestion. According to the suggestion raised by the reviewer, we have added and modified some contents and shown as the follows:

Some contents have been added in the part of introduction and shown as the follows:

(1) The increasing demand for the limited forest resources in various applications, has led to the shortage in wood supply [1]. Thus, there is an urgent need to look for new materials as alternatives of wood. Bamboo is one of the fastest growing natural plants in the world, which grows to its maximum height in about 3 months and reaches maturity in 3-4 years [2], well-exceeding the 20–60 years growth cycle of traditional timber used in structural applications.

(2) Out of many natural materials, bamboo offers significant structural and environmental advantages given its rapid growth, moisture resistance, climate tolerance, and tensile strength comparable to mild steel, good toughness, low processing cost, biodegradability, and so on [4].

(3) Therefore, bamboo and bamboo products have been widely used in the fields of decoration, architecture, furniture, gardens as alternatives of wood. [5-8], and there is a momentum of rapid growth. However, bamboo is easier to mildew because it contains more sugar, starch, protein and so on than wood,...

Some contents have been modified, such as:

(1) When the concentration of citral was 100 mg/ml, the diameters of the inhibitory zone of PC, TV, AN, and Hun were 31.57, 32.08, 36.64, and 30.42 mm, respectively.

(2) The concentration unit is “mg/ml”.

(3) We have adjusted the font size in the figures:

Figure 5. Effects of citral on absorbance at 260nm of extracellular fluids of bamboo mold (0: control group; MIC: minimal inhibitory concentration treated group; MBC: minimal bactericidal concentration treated group; PC, TV, AN, Hun were *Penicillium citrinum*, *Trichoderma viride*, *Aspergillus niger* and a hybrid fungi group comprising PC, TV, and AN, respectively).

Figure 6. Effect of citral on pH value of extracellular fluid of bamboo mold(0: control group; MIC: minimal inhibitory concentration treated group; MBC: minimal bactericidal concentration treated group; PC, TV, AN, Hun were *Penicillium citrinum*, *Trichoderma viride*, *Aspergillus niger* and a hybrid fungi group comprising PC, TV, and AN , respectively).

(4) We have adjusted the format of our references. Such as:

1.F. Bohlin, A. Roos. 2006. Wood fuel supply as a function of forest owner preferences and management styles. *Biomass Bioenergy* 22, (2002) 237-249. (doi:10.1016/S0961-9534(02)00002-8)

2. J. Scurlock, D. Dayton, B. Hames. 2000. Bamboo: an overlooked biomass resource?. *Biomass Bioenergy* 19, 229–244. (doi:10.1016/S0961-9534(00)00038-6)

3. Zhou B, Fu M, Yang X, Xie J, Li Z. 2006. Energy-oriented bamboo species resources and potential for exploitation. *World Forestry Research* 19, 49-52. (doi: 10.3969/j.issn.1001-4241.2006.06.011)

.....

5.Figures are missing explanations of the abbreviation and figure captions are short.

Response: Thanks to the reviewer for your comments. According to the suggestion raised by the reviewer, they have been modified, as follows:

Figure 1. Inhibitory effect of citral against bamboo mold (PC、TV、AN、Hun were *Penicillium citrinum*, *Trichoderma viride*, *Aspergillus niger* and a hybrid fungi group comprising PC, TV, and AN , respectively).

Figure 2. Effects of citral on the diameters of the inhibition zones of bamboo mold (PC、TV、AN、Hun were *Penicillium citrinum*, *Trichoderma viride*, *Aspergillus niger* and a hybrid fungi group comprising PC, TV, and AN , respectively).

Figure 3. Effect of citral on mycelia morphology of bamboo mold (a₁, a₂, and a₃ were PC control group, minimal inhibitory concentration treatment group and minimal bactericidal concentration treatment group, respectively; b₁, b₂, and b₃ were the TV respectively control group, minimal inhibitory concentration treatment group and minimal bactericidal concentration treatment group ; c₁, c₂, and c₃ were the AN control group, minimal inhibitory concentration treatment group and minimal bactericidal concentration treatment group , respectively).

Figure 4. Effects of citral on cellular structures of bamboo mold (a₁, a₂, and a₃ were PC control group, minimal inhibitory concentration treatment group and minimal bactericidal concentration treatment group, respectively; b₁, b₂, and b₃ were the TV respectively control group, minimal inhibitory concentration treatment group and minimal bactericidal concentration treatment group ; c₁, c₂, and c₃ were the AN control group, minimal inhibitory concentration treatment group and minimal bactericidal concentration treatment group , respectively).

Figure 5. Effects of citral on absorbance at 260nm of extracellular fluids of bamboo mold (0: control group; MIC: minimal inhibitory concentration treated group; MBC: minimal bactericidal concentration treated group; PC, TV, AN, Hun were *Penicillium citrinum*, *Trichoderma viride*, *Aspergillus niger* and a hybrid fungi group comprising PC, TV, and AN, respectively).

Figure 6. Effect of citral on pH value of extracellular fluid of bamboo mold(0: control group; MIC: minimal inhibitory concentration treated group; MBC: minimal bactericidal concentration treated group; PC, TV, AN, Hun were *Penicillium citrinum*, *Trichoderma viride*, *Aspergillus niger* and a hybrid fungi group comprising PC, TV, and AN, respectively).

Figure 7. The anti-mildew photographs of bamboo strips treated with citral at different concentrations on the 28th day(PC, TV, AN, Hun were *Penicillium citrinum*, *Trichoderma viride*, *Aspergillus niger* and a hybrid fungi group comprising PC, TV, and AN, respectively).

Editors

Response: Thanks for your kind reminder. We have ensured that all equations included in the paper are editable text. For example:

Bacteriostatic rate = (diameter of inhibition zone in treatment group – diameter of inhibition zone in control group)/diameter of inhibition zone in control group × 100% -----(1)

$$E = (1 - \frac{D_1}{D_0}) \times 100\% \text{----- (2)}$$

Response: Thanks for your kind reminder and suggestion. According to your suggestion and the guidelines at <https://royalsociety.org/journals/ethics-policies/openness/>, we have modified it and shown as the follow:

J.Z. carried out the work, participated in data analysis, participated in the design of the study and drafted the manuscript; C.D. conceived of the study, designed the study and critically revised the manuscript; Q.L., A.H., and R.P. helped to perform the experiments and analyse part date; F.S., and W.Z. provided experimental guidance. All authors gave final approval for publication and agree to be held accountable for the work performed therein.

Response: Thanks for your kind suggestion. We have adjusted the format of our references. Such as:

1. F. Bohlin, A. Roos. 2006. Wood fuel supply as a function of forest owner preferences and management styles. *Biomass Bioenergy* **22**, (2002) 237-249. (doi:10.1016/S0961-9534(02)00002-8)

2. J. Scurlock, D. Dayton, B. Hames. 2000. Bamboo: an overlooked biomass resource?. *Biomass Bioenergy* **19**, 229–244. (doi:10.1016/S0961-9534(00)00038-6)

3. Zhou B, Fu M, Yang X, Xie J, Li Z. 2006. Energy-oriented bamboo species resources and potential for exploitation. *World Forestry Research* **19**, 49-52. (doi: 10.3969/j.issn.1001-4241.2006.06.011)

.....